# A Bridge-Shaped Vibration Energy Harvester with Resonance Frequency Tunability under DC Bias Electric Field

**DOI:** 10.3390/mi13081227

**Published:** 2022-07-31

**Authors:** Guan Duan, Yingwei Li, Chi Tan

**Affiliations:** 1School of Chemistry and Civil Engineering, Shaoguan University, Shaoguan 512005, China; duanguansgu@sgu.edu.cn; 2School of Civil Engineering and State Key Laboratory of Water Resources and Hydropower Engineering Science, Wuhan University, Wuhan 430072, China; t_tanchi@126.com; 3School of Intelligent Construction, Wuchang University of Technology, Wuhan 430223, China; 4Wuhan Dislocation Technology Company, Wuhan 430072, China

**Keywords:** bridge-shaped energy harvester, piezoelectric bimorph, resonance frequency, direct current electric field, output power

## Abstract

A vibration piezoelectric energy harvester (PEH) is usually designed with a resonance frequency at the external excitation frequency for higher energy conversion efficiency. Here, we proposed a bridge-shaped PEH capable of tuning its resonance frequency by applying a direct current (DC) electric field on piezoelectric elements. A theoretical model of the relationship between the resonance frequency and DC electric field was first established. Then, a verification experiment was carried out and the results revealed that the resonance frequency of the PEH can be tuned by applying a DC electric field to it. In the absence of an axial preload, the resonance frequency of the PEH can be changed by about 18.7 Hz under the DC electric field range from −0.25 kV/mm to 0.25 kV/mm. With an axial preload of 5 N and 10 N, the resonance frequency bandwidth of the PEH can be tuned to about 13.4 Hz and 11.2 Hz, respectively. Further experimental results indicate that the output power and charging response of the PEH can also be significantly enhanced under a DC electric field when the excitation frequency deviates from the resonance frequency.

## 1. Introduction

With the increasing application of low-power consumption electronics and the Internet of Things (IoT), developing energy harvesters that can convert ambient energy into electricity to realize self-power for micro-devices has been the focus of many research efforts in past decades [1,2,3]. These energy harvesters can be designed to scavenge energy and work as long-lasting power for small-scale electronic devices, such as wearable and distributed sensors. Moreover, they can reduce disposal pollution [4,5,6] and replace batteries that require regular maintenance.

A lot of progress has been made in harvesting different kinds of ambient energy, e.g., mechanical vibration [7,8,9,10], wind blow [11,12], fluid flow [13], raindrop impact [14], and even human body motion [15,16]. Several kinds of small-scale generators have been developed accordingly, such as electromagnetic generators [17], triboelectric nanogenerators [18], piezoelectric generators [19], and ferroelectric generators [20]. Among these aforementioned reports, most attentions avert to PEHs, especially vibration PEHs, due to piezoelectric materials having many advantages, such as good electromechanical coupling characteristics, higher energy density, simpler architectures, and easier integration with MEMS [21]. These advantages make them more feasible for use in self-powered micro-devices.

However, the excitation frequency from the environment is not stable and varies in a wide frequency range, while the most common PEHs typically operate near the resonance frequency to achieve a higher output response [22]. For some structures, the resonance frequency changes as an increase in serving time for the loss of constraint or characteristic change of material. Additionally, for some devices, e.g., piezoelectric resonators, temperature fluctuation can lead to a deviation of the resonance frequency which can only be tuned by the application of a DC bias field for frequency-temperature compensation [23,24,25]. These obstacles limit the application of PEHs in self-powered micro-devices.

To enhance the performance of PEHs, many researchers focus on broadening the bandwidth of PEHs [2]. Several strategies have been developed, including tuning the resonance frequency by axial load [26], a multi-degree-of-freedom (multi-DOF) harvesting mechanism [27], a mono-stable [28], bi-stable nonlinear mechanism [29], and frequency-up-conversion mechanism [30]. Nevertheless, the structure of the PEHs mentioned above cannot be changed quickly and flexibly along with the random fluctuation of ambient excitation frequency, especially in some packaged devices. Fortunately, as mentioned above, some reports have implied that DC bias can influence the frequency of piezoelectric resonators [23,24,25]. In our previous work [31], under a pre-loading DC electric field of −0.5 to 0.75 kV/mm, the resonance frequency of a piezoelectric cantilever energy harvester can be tuned, but with a limited tunable frequency range of 12 Hz.

In this work, a new bridge-shaped PEH with a constrained double-end was proposed to further improve the bandwidth of the resonance frequency by tuning the DC electric field to piezoelectric elements. First, the theoretical model of the relationship between the DC electric field and the resonance frequency was deduced, and the feasibility of our resonance frequency tuning strategy was demonstrated by calculation and a simple test. Then, the experiments were carried out to successfully verify the tunability of the resonance frequency for the PEH under a different axial preload and DC electric field. At last, the performance of the proposed PEH is further investigated by experiments, such as the response voltage against different accelerations and load resistance, and the power output under load resistance and charging response of the PEH under a DC electric field.

## 2. The Theoretical Model and Design Principle

### 2.1. The Resonance Frequency of a Simply Supported Bimorph without Axial Preload

The key part of our proposed PEH with a capacity of resonance frequency tunability by DC electric field is shown in Figure 1a. The structure of the designed bimorph consists of two PZT layers and a brass layer. The detailed geometric and physical variables are given in Table 1. According to the theory of Vibration, the resonance frequency of a bimorph can be expressed as follows [26]:(1)$$f_{0}=\frac{1}{2\pi }\sqrt{\frac{k_{0}}{m_{0}}}$$
where $m_{0}$ is the effective mass acting on the center of the beam; it can be calculated by
(2)$$m_{0}=\frac{bL}{2}\left(\rho_{c}h_{c}+2\rho_{p}h_{p}\right)$$
where $k_{p}$ is the transverse stiffness of the center of the bimorph. Considering the crispness of the PZT layer, only the brass layer is screwed on the base. As the thickness of our brass layer is far less than the width and length, the designed bimorph can be simplified as a simply supported bimorph. According to the Euler–Bernoulli beam model, the transverse stiffness of a homogeneous simply supported bimorph can be expressed as follows:(3)$$k_{0}=EI{\left(\frac{\pi }{L}\right)}^{4}\left(\frac{L}{2}\right)$$
where *EI* represents the bending stiffness, which can be given by
(4)$$EI=E_{c}\frac{h_{b}^{3}}{12}+2E_{p}\left(\frac{h_{p}^{3}}{12}+h_{p}\left(\frac{h_{p}+h_{c}}{2}\right)\right)$$

### 2.2. The Resonance Frequency of a Simply Supported Bimorph with Axial Preload 

The resonance frequency of a simply supported bimorph will change as its stiffness changes with axial preload. In our proposed model, the axial preload *P* will enhance the stiffness of a simply supported bimorph. After applying an axial preload, the stiffness of the bimorph $k_{app}$ is defined as
(5)$$k_{app}=k_{0}+k_{p}$$
where $k_{p}$ is the stiffness induced by axial tensile preload *P*; then, $k_{p}$ is defined as [32]
(6)$$k_{p}=P{\left(\frac{\pi }{L}\right)}^{2}\left(\frac{L}{2}\right)$$

Figure 1b shows the curves of the resonance frequency vs. axial tensile load changing from 0 N to 10 N; the test results are consistent with the theoretical results calculated by Equation (6). This inspires us to find a way to tune the resonance frequency of our proposed PEH by changing the axial preload, which can be implemented only by tightening or loosening the screw on the designed structure manually. It can be seen that there is a difference between the theoretical and the measured resonant frequencies. This difference is mainly due to load sensor measurement error and mechanical assembly error, and the real tensile load value is larger than the measured displayed value, but the overall trend of the two lines is very similar.

### 2.3. The Resonance Frequency of Simply Supported Bimorph with Both Axial Preload and DC Electric Field

The application of an electric field on a piezoelectric material can cause deformation, which is called electric-induced strain. As shown in Figure 1a, when the DC electric field is applied along the polarization direction, the transverse and longitudinal deformation of the PZT layer will induce a tensile or compressive force in the bimorph when it is constrained on both ends. Then, it is concluded that the DC electric field can be used to tune the resonance frequency of the bimorph.

For a single PZT layer in Figure 1a, the relationship among axial preload $X_{1}$, longitudinal strain $x_{1},$ and DC electric field $E_{3}$ can be expressed in the form of simplified piezoelectric constitutive equations as follows:(7)$$x_{1}=s_{11}^{E}X_{1}+d_{31}E_{3}$$
(8)$$D_{3}=d_{31}X_{1}+\varepsilon_{33}^{T}E_{3}$$

It can be seen from Figure 1a that the longitudinal deformation of two PZT layers and the brass layer should be the same so that the variation of axial preload in bimorph can be given by
(9)$$\Delta F_{E}=\frac{d_{31}E_{3}}{\left(2/bE_{c}h_{c}+s_{11}^{E}/bh_{p}\right)}$$

Furthermore, the transverse stiffness of the load sensor is much less than the steel base, so the transverse displacement of the sensor should coordinate with the longitudinal deformation of the bimorph. Then, the variation of axial preload in bimorph is modified to
(10)$$\Delta F_{Es}=\frac{2\Delta F_{E}L}{E_{c}A_{c}\left(1/k_{s}+L/E_{0}A_{0}\right)}$$
where $E_{0}$ and $A_{0}$ are the effective Young’s modulus and section area of bimorph, respectively, and they are defined as
(11)$$E_{0}=\frac{2E_{p}h_{p}+E_{c}h_{c}}{2h_{p}+h_{c}}$$
(12)$$A_{0}=2h_{p}+h_{c}$$

Therefore, when the axial preload and DC electric field $E_{3}$ are applied on the bimorph, the axial preload $P_{E}$ can be described as
(13)$$P_{E}=P+\Delta F_{Es}$$

Finally, considering the comprehensive influence of the axial preload, DC electric field, and stiffness of the load sensor, the ultimate form of the stiffness and resonance frequency of the PEH are given by
(14)$$k_{PE}=P_{E}{\left(\frac{\pi }{L}\right)}^{2}\left(\frac{L}{2}\right)$$
(15)$$f_{E}=\frac{1}{2\pi }\sqrt{\frac{k_{0}+k_{PE}}{m_{0}}}$$

It should be noted that our theoretical model does not include the damping effect and geometrical nonlinearity which makes the theoretical model more complicated. Based on the electromechanical coupling, considering the coordination of displacement between the sensor, PZT layer, and brass layer, we deduced the relationship between the resonance frequency and the DC electric field. That is the principle behind tuning the resonance frequency of a bridge-shaped PEH by a DC electric field under an external driving force. Based on the above, the following experiment was carried out.

## 3. Experimental Validations and Discussion

### 3.1. Prototype Fabrication and Experimental Setup

In order to realize the capability of tuning the resonance frequency with a DC electric field, we designed and fabricated a prototype, as shown in Figure 2a, and conducted experiments to verify the performance in multiple perspectives, e.g., output voltage and power, load resistance, and charging response. An L-type steel structure is combined with a Single point load sensor, by which the transverse load can be measured and tuned by a screw quickly. The stiffness of the structure is specially designed with enough stiffness to avoid the displacement error made by the lack of stiffness.

A brass layer is sandwiched and glued with two PZT-5A piezoelectric layers (manufactured by Shenzhen FR Electroacoustic Technology Co, LTD, Shenzhen, China), and a small amount of conductive silver glue was used to make sure the glued electrode and brass layer were conductive. The poling direction of the two PZT layers mainly working in $d_{31}$ mode is the same, and marked by a red arrow in Figure 2a, to ensure that the two PZT layers are connected in parallel. The material parameters of PZT and brass are listed in Table 1. Then, the beam was clamped to the L-type structure and the Single point load cell through pressure and friction. Small Al_2_O_3_ ceramic plates were interposed between brass plates and clamps for insulation. Special attention must be paid to the experimental temperature since temperature fluctuations will cause significant stretching and tension of piezoelectric beams [33] which will affect the resonance frequency of the PEH. The ambient temperature in this experiment is kept constant at 20 °C during the experiment to ensure better experimental accuracy.

According to the schematic diagram in Figure 2a, the experimental device layout is shown in Figure 2b. The combined PEH structure of the bridge is connected to the shaker HEV-200 (Nanjing University of Aeronautics and Astronautics, Nanjing, China) by bolts, and the acceleration sensor is fixed on the L-shaped steel structure as well. The shaker is connected to a power amplifier and a signal generator. After the acceleration sensor is connected with the charge amplifier, the voltage signal is output to the oscilloscope to measure the acceleration. The upper and lower electrodes of the two piezoelectric plates are connected together to form an output terminal, and the middle electrode becomes another output terminal. The two output terminals are connected to the rectifier circuit and the output voltage and power. Before being connected to the rectifier circuit, two 0.47 μF capacitors are used to cut off the DC power supply and allow the alternating current to pass through.

### 3.2. Experimental Results and Discussion

#### 3.2.1. The Output Voltage under DC Electric Fields

We first investigated the resonance frequency variation of the PEH under different axial preloads and DC electric fields. The output open-circuit voltage as a function of frequency under different axial preloads and DC electric fields are plotted in Figure 3a. Under an acceleration of 0.5 g and an axial preload set at 0 N, 5 N, and 10 N, the floating range of the resonance frequency of the PEH is 18.7 Hz (217.3 Hz to 236 Hz), 13.4 Hz (259.1 Hz to 272.5 Hz), and 11.2 Hz (293 Hz to 304.2 Hz), respectively, when the electric field applied to the PEH was set to −0.25 kV/mm, −0.125 kV/mm, 0, 0.125 kV/mm, and 0.25 kV/mm. It is obvious that once the external excitation frequency deviates from the resonance frequency, the output voltage drops sharply. That is to say, the proposed strategy for tuning the resonance frequency with a DC electric field can make the PEH run at a resonance frequency in bandwidth from 11.2 to 18.7 Hz, in which the output voltage and power will keep at a high value and the energy conversion efficiency will be greatly improved.

Figure 3b presents a comparison of the measured resonance frequency (bold line), measured axial preload (dash line), and the theoretical resonance frequency (thin line) under different DC electric fields. Under an axial preload set at 0 N, 5 N, and 10 N, the theoretical floating range of resonance frequency of the PEH is 18.4 Hz (200.5 Hz to 218.9 Hz), 15.2 Hz (246.4 Hz to 261.6 Hz), 13.3 Hz (285 Hz to 298.3 Hz), respectively, when the DC electric field varies from −0.25 kV/mm to 0.25 kV/mm.

It can also be seen from Figure 3b that both the measured and the theoretical resonance frequency of the PEH increase as the axial preload increases, and the higher the preload, the smaller the resonance frequency tunable range. This phenomenon agrees with the previous theoretical verification.

#### 3.2.2. The Output Voltage under Different Accelerations and Load Resistances

Figure 4a illustrates the frequency dependence of the output voltage under ambient vibration excitation with different accelerations. As observed in Figure 4a, with an axial preload of 1.2 N and DC electric field set to 0 kV/mm, the open-circuit peak-peak voltage of the PEH is 8 V, 11.8 V, and 15 V, respectively, when the vibration acceleration is 0.5 g, 1 g, 1.5 g. The proposed PEH shows a satisfactory voltage response in a low vibration acceleration of 0.5 g, and the maximum output voltage occurs at its resonance frequency of about 232.3 Hz.

Figure 4b shows the output response voltage of the PEH with various load resistances. It can be seen that with an axial preload of 1.2 N and in the absence of a DC electric field, the voltage and the resonance frequency increase gradually as the load resistance increases. The output voltage $V_{p-p}$ across the load resistance $R_{L}$ is 0.68 V, 2.56 V, 4.6 V, and 8 V, when the load resistance is 1 kΩ, 8 kΩ, 16 kΩ, and ∞ (open-circuit) and the resonance frequency range is from 224 Hz to 232.3 Hz.

#### 3.2.3. The Output Power of PEH with Various Load Resistance

In order to achieve maximum output voltage and power for higher energy conversion efficiency, not only the resonance frequency but also the load resistance should be tuned. Figure 5 plots and compares the output power of the PEH as a function of load resistance under different DC electric fields, with an axial preload of 5.6 N and vibration acceleration of 0.5 g, where the output power is calculated by $P=V_{p-p}^{2}/\left(2R_{L}\right)$. It can be seen that the power of the PEH first increases and then decreases with the increasing load resistance $R_{L}$, and reaches the maximum value of 0.46 mW at $R_{L}\;$= 16 kΩ, when the PEH works at the resonance frequency of 270.2 Hz without a DC electric field (black line). With a DC electric field of −0.25 kV/mm and 0.25 kV/mm, the new resonance frequency is offset to 266.5 Hz and 281 Hz, meanwhile, the output power of the PEH reaches the maximum value of 0.453 mW and 0.385 mW, respectively. For comparison, when DC electric field is tuned off, the maximum power value measured at the excitation frequency of 266.5 Hz and 281 Hz (both are not resonance frequencies), apparently decreases to 0.418 mW and 0.137 mW. In other words, the output power of the PEH is enhanced by 180% and 8.3% after applying a DC electric field of −0.25 kV/mm and 0.25 kV/mm, respectively. It is evident that by tuning the resonance frequency of the PEH with a DC electric field, its power output can be significantly enhanced at a frequency deviating from its initial resonance frequency.

#### 3.2.4. The Charging Performance of PEH under DC Electric Field

Finally, we investigated the charging response with or without a DC electric field under an axial preload of 2.1 N and an acceleration of 0.5 g; a 33 μF capacitor is connected in parallel with a full wave rectifying bridge (shown in Figure 2a). The DC charging voltage across the capacitor is measured by an electrometer. Figure 6 illustrates the charging voltage as a function of time with or without a DC electric field. It can be seen that the charging voltage tends to saturate quickly, after nearly 10 s, but the saturation voltage varies in five different cases.

When the resonance frequency of the PEH is tuned to 244.3 Hz without a DC electric field, the saturation voltage across the capacitor reaches 2.02 V. After we applied a DC electric field of 0.25 kV/mm and −0.25 kV/mm to the PEH, the new tuned resonance frequency was 260.9 Hz and 241.4 Hz, respectively; meanwhile, the charging saturation voltage across the capacitor becomes 1.94 V and 1.79 V, which are smaller than 2.02 V. Then, when we turned off the DC electric field and tuned the excitation frequency to 260.9 Hz and 241.4 Hz (both are not resonance frequencies), the saturation voltage across the capacitor dropped to 0.2 V and 0.98 V. That is to say, the charging saturation voltage across the capacitor can be increased by 870% and 82.7%, respectively, under the DC electric fields of 0.25 kV/mm and −0.25 kV/mm when the excitation frequency deviates from the resonance frequency (244.3 Hz) by 16.6Hz and −2.9 Hz. The charging response results prove that the DC electric field not only enhances the output power but also the charging performance.

## 4. Conclusions

In summary, to broaden the effective working bandwidth of the PEH for improving its energy harvesting efficiency, a bridge-shaped PEH with resonance frequency tunability by applying a DC electric field on piezoelectric elements is designed and fabricated, and its performance is verified by theoretical analysis and experimental techniques.

Theoretical analysis reveals that the DC electric field is directly proportional to the longitudinal stiffness of the PEH, resulting in a change in the resonance frequency. The precise DC electric field needed for tuning the resonance frequency can be calculated by the established relationship between the resonance frequency and DC electric field. The experimental results on the resonance frequency tunability prove that applying a DC electric field from 0.25 kV/mm to −0.25 kV/mm on piezoelectric elements can induce a resonance frequency floating range of 11.2 Hz to 18.7 Hz when given an axial preload of 10N to 0N, accordingly. Further experiments were conducted on the output performance of the PEH. The results show that the output power of the PEH can be increased by 180%, and the charging saturation voltage across the capacitor can be increased up to 870% under a tuning DC electric field of 0.25 kV/mm.

All the results suggest that the proposed PEH has the potential application for special conditions such as when the PEH structure cannot be changed.

## Figures and Tables

**Figure 1 micromachines-13-01227-f001:**
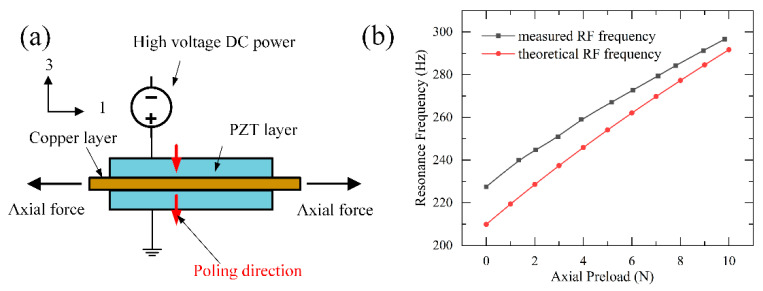
(**a**) Schematic of the bimorph in the proposed PEH; (**b**) The measured and theoretical resonance frequency versus preload.

**Figure 2 micromachines-13-01227-f002:**
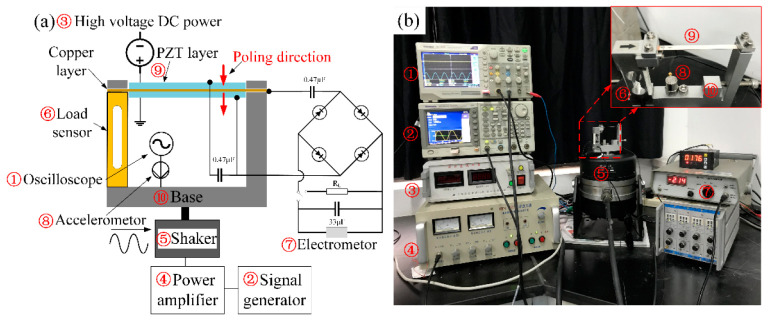
(**a**) Illustration of the measurement setup; (**b**) Photograph of the experimental setup.

**Figure 3 micromachines-13-01227-f003:**
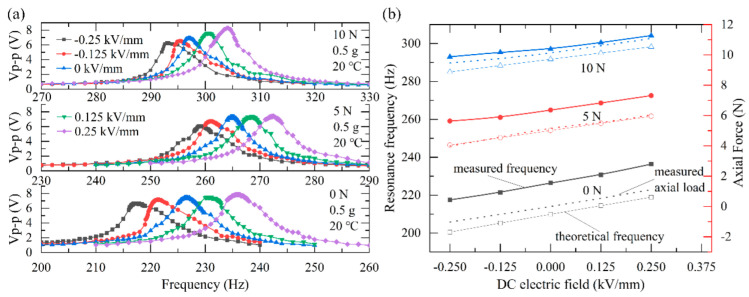
(**a**) The output voltage as a function of frequency under different DC electric fields and axial preload; (**b**) The measured resonance frequency, the measured axial preload, and the theoretical resonance frequency under different DC electric fields and different preloads.

**Figure 4 micromachines-13-01227-f004:**
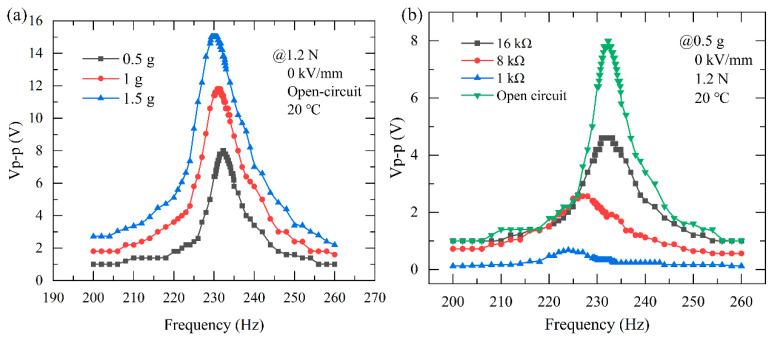
The output performance of the PEH. (**a**) Output response voltage as a function of frequency under different acceleration; (**b**) Output response voltage with various load resistances.

**Figure 5 micromachines-13-01227-f005:**
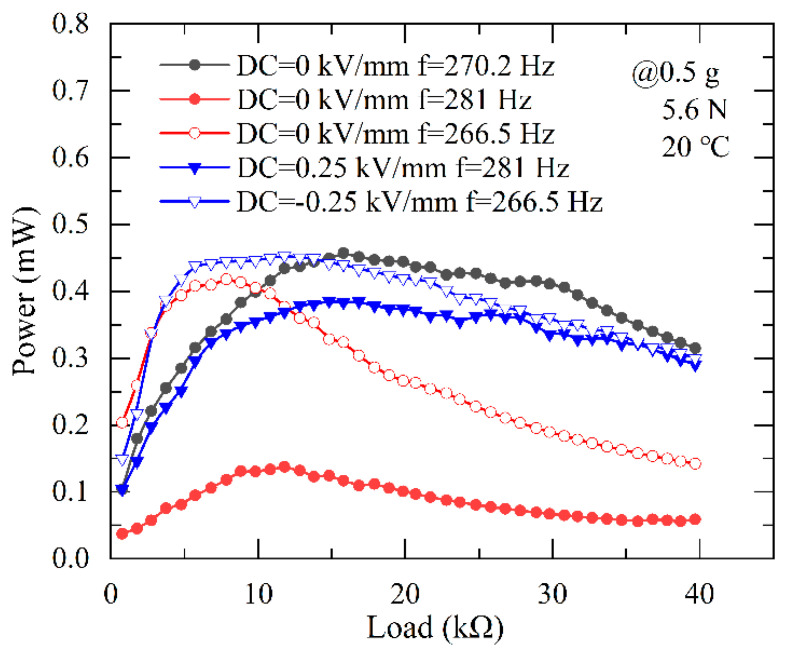
The output power tuning performance of the PEH under different DC electric fields.

**Figure 6 micromachines-13-01227-f006:**
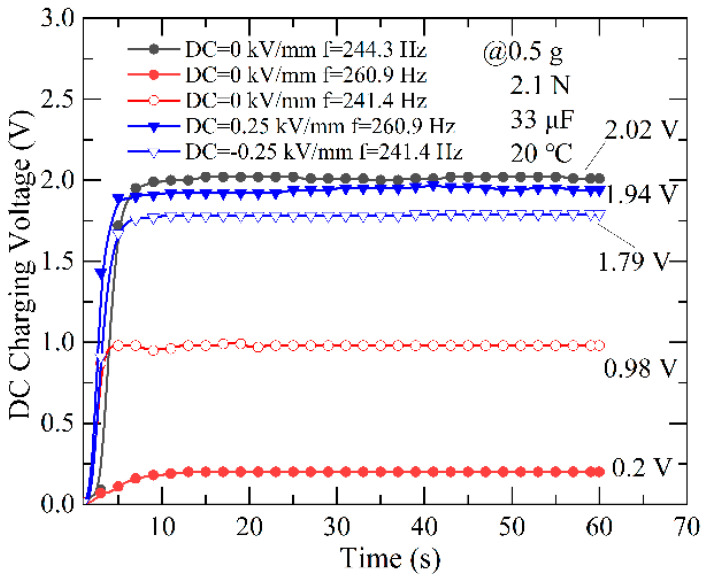
The charging response tuning performance of the PEH under different DC electric fields.

**Table 1 micromachines-13-01227-t001:** Variable descriptions and their values.

Symbol	Description	Value	Units
*b*	Bimorph width	5	mm
*L*	Bimorph length	56	mm
$h_{p}$	Piezo layer thickness	0.2	mm
$h_{c}$	Brass layer thickness	0.1	mm
$E_{p}$	Piezo Young’s modulus	65	GPa
$E_{c}$	Brass Young’s modulus	126	GPa
$\rho_{p}$	Piezo density	7750	kg/m^−3^
$\rho_{c}$	Brass density	7850	kg/m^−3^
$k_{s}$	Stiffness of load sensor	20	kg/mm
$s_{11}^{E}$	Elastic compliance	1.65 × 10^−11^	m^2^/N
$d_{31}$	Piezoelectric coefficient	2.74 × 10^−10^	C/N
g	Gravity acceleration	9.8	N/kg

## Data Availability

Data is contained within the article.

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
