# Peer review of "A Bridge-Shaped Vibration Energy Harvester with Resonance Frequency Tunability under DC Bias Electric Field"

_micromachines, 2022, doi:10.3390/mi13081227_

Round 1

Reviewer 1 Report

In the research article titled, "A bridge-shaped vibration energy harvester with resonance frequency tunability under DC bias electric field", the authors have designed piezoelectrical equations based on a piezoelectric energy harvester and investigated its materials, performance, and characteristics. With a large amount of theoretical and experimental tests, the device can achieve an optimal output. However, this design lacks the explanation of novelty. More efforts need to be poured into innovation miming and quality improvement. Therefore, the authors need to address all the raised issues before consideration of acceptance:

1. The abbreviation of RF is a prescriptive usage for the meaning of Radio Frequency (RF). The authors should change another abbreviation or use resonant frequency directly.

2. The language of the manuscript should be polished again for better explanation.

3. The manuscript lacks of the description of Figure 1(b), where the reason for the difference between the theoretical and measured resonant frequencies should be explained clearly.

4. Figure 5 demonstrates the power output under different DC fields varied from -0.25, 0, 0.25kV/mm. However, the data is insufficient without  corresponding frequencies of -0.25/266.5 and 0.25/281. If it is no need to add these two options, the author should give the reason. The same explanation should be added in Figure 6.

5. The bridge-shaped device works by adding electric fields on the PZT bimorph for energy harvesting from piezoelectric effect to tune or broaden resonant frequency, more detailed description of the novelty and application scenario of this paper should be provided. 

6. Suggest adding the following references that are closely relevant to this work: Advanced Science 2021, 8, 2101834. 10.1002/advs.202101834; Micromachines, 2022, 13(2), 00232. 10.3390/mi13020232.

Reviewer 2 Report

This work proposed a bridge-shaped PEH capable of tuning its RF by applying direct current electric (DC)  field on piezoelectric elements. A theoretical model of the relationship between the RF and DC electric field was first established. Then the verification experiment was carried out and the results 16 reveal that the RF of PEH can be tuned by applying DC electric field to it. This work can be considered for publication after the following comments:

1. Author should provide more detail in the introduction part on the Vibration piezoelectric energy harvester and its performance parameters in terms of device design, materials, and applications.

2. Author should discuss device optimization parameters and results.

3. Current can be compared in form of a table with an already reported Vibration piezoelectric energy harvester.

4. Application should provide the different applications of Vibration piezoelectric energy harvester. 
